# Physical Activity Trajectories among Persons of Turkish Descent Living in Germany—A Cohort Study

**DOI:** 10.3390/ijerph17176349

**Published:** 2020-08-31

**Authors:** Lilian Krist, Christina Dornquast, Thomas Reinhold, Heiko Becher, Katja Icke, Ina Danquah, Stefan N. Willich, Thomas Keil

**Affiliations:** 1Institute for Social Medicine, Epidemiology and Health Economics, Charité–Universitätsmedizin Berlin, Humboldt-Universität zu Berlin, Berlin Institute of Health, 10117 Berlin, Germany; thomas.reinhold@charite.de (T.R.); katja.icke@charite.de (K.I.); ina.danquah@uni-heidelberg.de (I.D.); stefan.willich@charite.de (S.N.W.); thomas.keil@charite.de (T.K.); 2German Center for Neurodegenerative Diseases (DZNE), 17489 Greifswald, Germany; christina.dornquast@dzne.de; 3Institute for Medical Biometry and Epidemiology, University Medical Center Hamburg-Eppendorf, 20251 Hamburg, Germany; h.becher@uke.de; 4Institute of Global Health (HIGH), Heidelberg University Hospital, 69120 Heidelberg, Germany; 5Institute for Clinical Epidemiology and Biometry, University of Würzburg, 97070 Würzburg, Germany; 6State Institute of Health, Bavarian Health and Food Safety Authority, 97688 Bad Kissingen, Germany

**Keywords:** physical activity, physical activity trajectories, migrants, cohort study

## Abstract

Physical activity (PA) behavior is increasingly described as trajectories taking changes over a longer period into account. Little is known, however, about predictors of those trajectories among migrant populations. Therefore, the aim of the present cohort study was to describe changes of PA over six years and to explore migration-related and other predictors for different PA trajectories in adults of Turkish descent living in Berlin. At baseline (2011/2012) and after six years, sociodemographics, health behavior, and medical information were assessed. Four PA trajectories were defined using data of weekly PA from baseline and follow-up: “inactive”, “decreasing”, “increasing”, and “stable active”. Multivariable regression analyses were performed in order to determine predictors for the “stable active” trajectory, and results were presented as adjusted odds ratios (aOR) with 95% confidence intervals (95%CI). In this analysis, 197 people (60.9% women, mean age ± standard deviation 49.9 ± 12.8 years) were included. A total of 77.7% were first-generation migrants, and 50.5% had Turkish citizenship. The four PA trajectories differed regarding citizenship, preferred questionnaire language, and marital status. “Stable active” trajectory membership was predicted by educational level (high vs. low: aOR 4.20, 95%CI [1.10; 16.00]), citizenship (German or dual vs. Turkish only: 3.60 [1.20; 10.86]), preferred questionnaire language (German vs. Turkish: 3.35 [1.05; 10.66]), and BMI (overweight vs. normal weight: 0.28 [0.08; 0.99]). In our study, migration-related factors only partially predicted trajectory membership, however, persons with citizenship of their country of origin and/or with poor language skills should be particularly considered when planning PA prevention programs.

## 1. Introduction

Physical activity (PA) contributes to a broad range of health benefits and is one of the most important protective factors for a variety of chronic diseases [1,2,3,4,5,6]. In later life, PA can prevent cognitive decline and improve physical functioning [7,8,9,10]. Among the elderly, activities of daily living can longer be maintained when engaging in regular PA [11,12]. Finally, the beneficial effects might delay premature mortality and increase one’s lifespan [6,13,14]. 

A common methodological approach to describe PA is its assessment at a single point in time [15]. However, this approach does not take into account the dynamic nature of PA. Thus, for a more detailed description of PA over a longer time period and a better understanding of its complexity, prospective trajectories including two or more time points are proposed [16,17]. PA trajectories can then be used to predict future health and physical functioning and are therefore a good measure for PA [5,9,11,15,18,19,20,21]. In addition, the identification of associated factors for specific PA trajectories is important for tailoring prevention efforts. 

Although the favorable effects of PA are widely proven and recognized, adherence to common recommendations is low in high-income countries including Germany, and PA tends to decline over one’s lifetime [22,23,24]. Many studies have been conducted to investigate reasons for this [25]. In a recent systematic review investigating PA trajectories, male sex, being Caucasian, non-smoking, having low television viewing time, higher socioeconomic status, no chronic illnesses, and family support for PA were positively associated with stable or increasing PA trajectories [17]. Another study showed associations of education, age, and employment status with PA trajectories [26]. 

In recent decades, predictors of PA have been described increasingly also in samples of ethnic minorities [27]. This is of great public health importance, since this population group is at special risk due to a high prevalence of overweight and obesity as well as chronic diseases such as diabetes [28]. A recent review and a resulting developed system-based framework reported that PA and sedentary behavior in ethnic minority groups in Europe were mostly associated with social and environmental factors such as sex, religion, and personal or cultural beliefs regarding PA [29,30]. Other studies that investigated associated factors with PA reported acculturation status, length of stay, own migration experience (first migration generation), and citizenship as migration-related factors as well as age, sex, socioeconomic status, employment status, education, psychosocial, and environmental factors [27,31,32,33]. A study from Australia described language skills and country of origin as predictive factors of PA assuming that the comprehension of prevention measures was easier for migrants from an English-speaking country [34]. A study among Turkish migrants in England and Germany concluded that age, sex, marital status, and the host country, but not acculturation were associated with PA [35]. 

While there is growing evidence for predictors of PA among migrants in general, longitudinal data on PA and predictors for PA trajectories in this population group are still scarce [33]. Since persons of Turkish descent form the largest group of persons with migration background in Berlin (6% of the Berlin population and 20% of all Berliners with migration background) [36], the aim of our study was to describe different trajectories of PA behavior as well as predictors of these trajectories in a prospective cohort study among persons of Turkish descent living in Berlin, Germany. 

## 2. Materials and Methods 

### 2.1. Study Design and Recruitment

For this cohort study, 557 adults of Turkish descent living in Berlin were recruited via a complex recruitment process (register-based and network approach) using an onomastic procedure to identify persons of Turkish descent. The recruitment strategies have been described by Reiss et al. in more detail [37]. Briefly, the baseline assessment (conducted between 2011 and 2012) was part of the pretest phase of the German National Cohort Study (NAKO) with the aim to evaluate different recruitment strategies among persons of Turkish descent [37,38]. All recruited participants completed a questionnaire and underwent some medical examinations (measurement of body height and weight, blood pressure, blood sample). After 6 years, all participants who had agreed to be re-contacted, received the follow-up questionnaire consisting of questions regarding health status, health behavior, health care utilization, and others. Baseline and follow-up recruitment were conducted using bilingual written invitations, as well as telephone contacts and home visits performed by bilingual study staff. A more detailed description of the follow-up recruitment has been provided by Krist et al. [39]. The study was approved by the ethical review committee of the Charité - Universitätsmedizin Berlin, Germany, and registered at the German Clinical Trials Register under the registration number DRKS00013545. Written informed consent was obtained from all participants.

### 2.2. Measures

#### 2.2.1. Physical Activity

Moderate-to-vigorous physical activity was assessed with two questions. 1. “On how many days per week are you physically active (any activity that increases your heart rate and makes you get out of breath) during an average week?”; participants could enter a number. 2. “How long are you physically active on average on those days when you sweat or are out of breath due to your physical activity?”. Response options for that question were “less than 10 min”, “10 to under 30 min”, “30 to under 60 min”, and “60 min or more”. PA minutes per week were calculated according to Krug et al. using the mean value of the answer categories for the question on duration and estimating the top category conservatively at 60 min [40]. Duration of weekly PA was then categorized into “inactive” (no activity), light PA (<150 min/week), moderate PA (150- <300 min/week), and high PA (≥300 min/week) in order to define the PA trajectories. Four PA trajectories were defined: “inactive”, if no activity was reported at baseline and follow-up; “decreasing”, if PA activity level was lower at follow-up than at baseline; “increasing”, if PA activity level was higher at follow-up than at baseline; and “stable active”, if PA activity level remained the same (at least light PA). More details are shown in Appendix A. 

#### 2.2.2. Socio-Demographics

As socio-demographic variables, we included sex (male/female), age (in years), marital status (married/not married) and educational level defined as years of attained formal education in Turkey and/or Germany (<10 years, 10–12 years, and >12 years), and net household income (EUR ≤ 1500 per month/EUR > 1500 per month). Age and household income were assessed at baseline and follow-up, all other variables were assessed only at baseline. 

#### 2.2.3. Migration-Related Factors

Three migration-related variables were included. Firstly, migration generation was compiled using information about the country of birth and the question whether participants had lived in Germany since birth. Participants who were born in Turkey or another country were categorized into the group with their own migration experience (1st generation), while participants who were born in Germany were defined as the group without migration experience (2nd generation). The second variable was citizenship, which was dichotomized into Turkish, if it was only Turkish, and into German, if it was German and Turkish or German alone. As a third variable, the chosen language of the questionnaire (German or Turkish) was included, since the participants’ choice reflects their language skills better than information about their mother tongue.

#### 2.2.4. Health Behavior and Diseases

At baseline, trained study personnel measured body height to the nearest 0.1 cm and body weight to the nearest 0.1 kg using a calibrated integrated measurement station (SECA model 764, Seca^®^, Hamburg, Germany). Body mass index (BMI) was calculated from these measurements as weight over height squared in kg/m^2^, and categorized into normal weight (BMI 18.5 to <25.0kg/m^2^), overweight (BMI 25.0 to <30.0kg/m^2^), and obesity (BMI ≥30.0kg/m^2^). BMI at follow-up was calculated using measured height at baseline and self-reported weight at follow-up. Smoking status was assessed at baseline and follow-up, and categorized into smoker (regular smoking), ex-smoker, and never-smoker. Lastly, lifetime prevalence of hypertension, diabetes, and dyslipidemia (self-report of physician’s diagnosis), assessed at baseline and follow-up, was considered.

### 2.3. Statistical Analyses

We considered our statistical approach as explorative rather than strictly hypotheses testing. For the definition of PA trajectories, we used the variables “weekly PA baseline” and “weekly PA follow-up” and defined four groups: “inactive”, “decreasing”, “increasing”, and “stable active”. A detailed description of the four trajectories is presented in the Methods section and in Appendix A. The trajectories were analyzed using descriptive methods of means and standard deviations for continuous data and absolute and relative frequencies for categorical data. Differences in characteristics by trajectory were tested with chi-square test for categorical variables and ANOVA for continuous variables. A multivariable logistic regression analysis was conducted to investigate associations between baseline variables (exposure) and “stable active” trajectory membership (outcome). Results of the multivariable regression analysis were presented as odds ratios (OR) with 95% confidence intervals (CI). The analyses were performed using SPSS Statistics for Windows (25.0.0.1, IBM Corp., Armonk, NY, USA). Creation of Sankey diagrams was performed using displayr.com (Displayr, Sydney, Australia).

## 3. Results

### 3.1. Characteristics of the Study Sample and PA Trajectories

Of the 557 baseline participants, 249 completed the follow-up questionnaire (further referred to as “participants”); the group of “non-participants” consists of 248 persons who refused to the follow-up actively or passively and 60 persons who could not be re-contacted. Among the non-participants, 247 had complete PA baseline data. Among the participants, 217 had complete baseline data, 220 had complete follow-up data; 197 had complete PA data for both baseline and follow-up and were included in the present analyses. Mean age of the sample was 49.9 ± 12.8 years at follow-up, 60.9% were women, 77.7% were first-generation migrants, and 50.5% had Turkish citizenship. 

According to the definition of PA trajectories, 23.7% showed an inactive trajectory, 12.2% a decreasing, 42.1% an increasing, and 19.3% a stable active trajectory. When comparing the four trajectories, we found differences for citizenship, language of questionnaire, and marital status (all *p*-values < 0.05). Participants in the inactive trajectory were older, more often men, more often had their own migration experience, preferred Turkish as the questionnaire language, and were more often married than the total sample. The decreasing PA trajectory contained mostly women, participants with low educational level and low household income, who preferred more often German as the language, more often had German citizenship, were more often not married, and had high hypertension prevalence and incidence. Participants in the increasing PA trajectory had mostly Turkish citizenship, high BMI, and were smokers. The stable active trajectory contained mostly participants with a high educational level and high net income, with German or dual citizenship and German as the preferred questionnaire language, as well as the most participants with normal weight. More details of the trajectory characteristics are shown in Table 1.

### 3.2. PA Trajectories among Participants and Non-Participants

Among both participants and non-participants, over half of the subjects were inactive at baseline (52% among participants, 63% among non-participants), and about 30% reported light PA, while only 12% and 8% reported moderate or high PA, respectively. Overall, PA increased among the participants over the period of six years, whereas 32% were reporting no activity at all at follow-up. PA trajectories of participants and non-participants are shown in Figure 1a,b, respectively. 

Figure 2 shows the four PA trajectories of the participants.

### 3.3. Predictors of PA Trajectories 

The results of the univariable and multivariable logistic regression analyses for potential determinants of stable active trajectory membership are presented in Table 2. Predictors for being in the stable active trajectory were a high educational level (OR 4.20, CI [1.10; 16.00]; *p* = 0.036 for more than 12 years of education compared to less than 10 years). Among the migration-related factors, German or dual citizenship (OR 3.60 [1.20; 10.86]; *p* = 0.023) and German as the preferred questionnaire language (OR 3.35 [1.05; 10.66], *p* = 0.041) were positively associated with being in the stable active trajectory. As the only health-related factor, a high BMI was negatively associated with trajectory membership. Compared to normal weight, participants with overweight had 72% lower odds and participants with obesity 71% lower odds (OR 0.28 [0.08; 0.99]; *p* = 0.047 and 0.29 [0.08; 1.08], *p* = 0.066), respectively) for being in the active stable trajectory. 

## 4. Discussion

### 4.1. Main Study Findings and Implications

In this study, four physical activity trajectories (“inactive”, “decreasing”, “increasing”, and “stable active”) were identified among a sample of adults of Turkish descent, living in Berlin. Similar results were reported by Pan et al. who described the same PA trajectories in a Taiwanese sample [26]. Similar trajectories (always sedentary, fast declining, stable moderate, and always active) were reported in a study from the U.S. among a sample of older women aged 70–79 years, however, the authors found no increasing trajectory which was explained with the high age of the study participants [41]. Most studies describe three to four trajectories or patterns [19,26,41], and less often more are reported, as in a European study conducted in Spain with more than 1600 participants, reporting very diversified patterns (high PA—consistent, moderate PA—mildly decreasing, low PA—increasing, moderate PA—consistent, and low PA—decreasing) [12].

The trajectories in our study sample showed differences regarding citizenship, preferred language of the questionnaire, and marital status. Subsequent multivariable regression analyses revealed that being in the stable active trajectory was predicted by only one of the included socio-demographic factors, namely high education (more than 12 years compared to less than 10 years). This is partially in line with several other studies showing that male sex, lower age, high education, and/or a high household income were predictors for favorable PA trajectories (increasing or persistently active) [17,21,33,42,43,44]. 

Among the migration-related factors, German or dual citizenship, and German as the preferred questionnaire language were associated with being in the stable active trajectory, while migration generation was not associated. The association with citizenship in our study is in agreement with a previous study from the U.S. showing higher PA among migrants with U.S. citizenship [45]. A recent study from the EU also showed significantly lower PA levels among nationals of non-EU countries in Austria compared to Austrian nationals [46]. Participants who chose German as the questionnaire language were more likely to be in the stable active trajectory, which is supported by a systematic review, including a wide range of countries, showing that poor knowledge of the language of the host country was claimed as having a large impact on PA as a barrier to get access to or understand PA recommendations [29]. Migration generation was not associated with being in the stable active trajectory. Similar results were reported by Koca et al. who investigated PA behavior in Turkish migrants in England and Germany and showed that migration generation and length of stay were not associated with PA. In contrast to our study however, language proficiency was not associated with PA, either [35]. 

While several studies showed that health-related factors like smoking or high BMI were associated with poor PA outcomes [21,47,48,49,50], the results of our study were not that concise. In our study, only overweight and obesity compared to normal weight subjects had decreased odds for being in the stable active trajectory, whereas smoking or having a cardiovascular risk factor were not associated with the PA trajectory. Since the overall prevalence of smokers was fairly high in our sample, this could have attenuated a possible effect of smoking. 

Most studies investigating relationships between PA and cardiovascular risk factors focus on PA as exposure, not as an outcome. Common results are that PA leads to a decreased risk of developing cardiovascular risk factors [51,52]. Our results did not show an association of cardiovascular risk factors with future PA, however, the importance of these factors as predictors or mediators for PA should be emphasized. Future studies should include these risk factors as exposure variables when investigating PA trajectories.

### 4.2. Strengths and Limitations

To our knowledge, this is the first study that investigated physical activity trajectories among a sample of adults with Turkish descent in Germany. A strength of the study was also the application of two different sampling approaches at baseline which led to the inclusion of persons who are generally difficult to reach for participation in epidemiologic studies as well as the intensive retention effort including home visits which contributed to the moderate recruitment success at follow-up. Furthermore, the bilingual questionnaire and staff in the recruitment office and during home visits may have contributed to more valid answers of the participants by reducing possible language or cultural barriers. However, certain limitations of our study need to be considered as well. Despite the moderate follow-up rate, the total sample size was rather small which may have reduced the statistical power of the study. A relevant selection bias is considered unlikely since baseline PA had a similar distribution in follow-up participants and in non-participants. Further, our analyses allow only the presentation of associations, but we cannot infer causal relationships. Another limitation is that PA was assessed only via self-report. Although the used questions for PA assessment were validated and are components of widely used questionnaires [40], future studies should use accelerometers or other means to objectively measure PA. Moreover, since PA was not the main focus in the used questionnaire, no information about different intensities and settings, nor sedentary behavior could be assessed. With the assessment of moderate-to-vigorous PA and the respective PA durations, we provide, however, the first results regarding PA trajectories for this population group. Future research might address those factors for a more detailed description of long-term PA in persons with a migration background. Lastly, more assessment points would have led to a more precise characterization of the respective PA trajectories. A third assessment is however foreseen.

## 5. Conclusions

In this study population with adults of Turkish descent in an urban German setting, only 20% of the participants remained physically active over several years. While high education, German citizenship, and good German language skills appeared to initiate healthy PA trajectories, own migration experience seemed to marginally determine motivation for PA. Therefore, linguistic peculiarities should be taken into account when designing culturally adapted lifestyle interventions, particularly for migrants who still hold their original citizenship.

## Figures and Tables

**Figure 1 ijerph-17-06349-f001:**
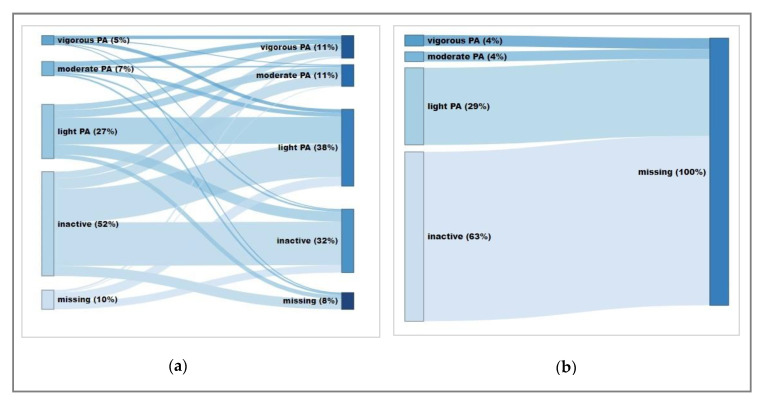
PA trajectories of (**a**) participants (study sample with PA data for at least one assessment point (*n* = 240)) and (**b**) non-participants with baseline PA data (*n* = 247). Inactive: no activity; light PA: <150 min/week; moderate PA: 150- <300 min/week; high PA: ≥300 min/week. Software: displayr.com (Displayr, Sydney, Australia).

**Figure 2 ijerph-17-06349-f002:**
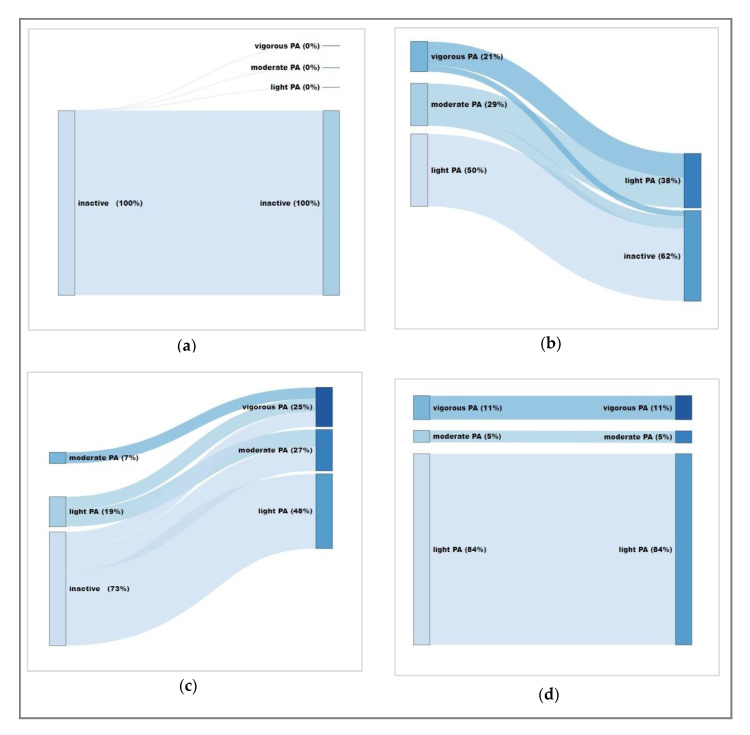
Weekly activity levels from baseline to follow-up for the (**a**) inactive (23.7%), (**b**) decreasing (12.2%), (**c**) increasing (42.1%), and (**d**) stable active (19.3%) trajectories. Inactive: no activity; light PA: <150 min/week; moderate PA: 150- <300 min/week; high PA: ≥300 min/week. Software: displayr.com (Displayr, Sydney, Australia).

**Table 1 ijerph-17-06349-t001:** Characteristics of total sample and physical activity (PA) trajectories.

	Physical Activity Trajectories										
Characteristics	Total *n* = 197		Inactive *n* = 52 (23.7%)		Decreasing *n* = 24 (12.2%)		Increasing *n* = 83 (42.1%)		Stable Active *n* = 38 (19.3%)		*p*-Value ^1^ (Baseline; Follow-up)
	baseline	follow-up	baseline	follow-up	baseline	follow-up	baseline	follow-up	baseline	follow-up	
	% or mean ± SD										
Time-stable (assessed only at baseline)											
Sex											0.275
Male	39.1		48.1		25.0		37.3		39.5		
Female	60.9		51.9		75.0		62.7		60.5		
Education											0.185
<10 years	36.5		36.5		41.7		37.3		31.6		
10–12 years	37.6		42.3		29.2		41.0		28.9		
>12 years	20.3		13.5		16.7		18.1		36.8		
Missing	5.6		7.7		12.5		3.6		2.6		
Migration generation											0.280
1st generation	77.7		81.6		62.5		78.5		80.6		
2nd generation	22.3		18.4		37.5		21.5		19.4		
Citizenship											0.034
Turkish	50.5		50.0		41.7		61.3		33.3		
German or dual	49.5		50.0		58.3		38.7		66.7		
Preferred language of questionnaire											0.007
Turkish	48.7		59.6		33.3		55.4		28.9		
German	49.5		40.4		66.7		44.6		71.1		
Married											0.032
yes	73.4		86.3		58.3		74.1		63.9		
no	26.6		13.7		41.7		25.9		36.1		
Time varying (assessed at baseline and follow-up)											
Age in years	43.7 ± 12.8	49.9 ± 12.8	45.0 ± 12.9	51.2 ± 12.9	44.1 ± 12.6	50.2 ± 12.5	43.3 ± 12.5	49.6 ± 12.4	42.5 ± 13.9	48.6 ± 14.0	b: 0.810; f: 0.815
Household net income											b: 0.513; f: 0.479
EUR ≤ 1500	29.4	32.6	34.6	29.4	37.5	45.8	26.5	31.7	23.7	30.6	
EUR > 1500	51.3	57.0	44.2	54.9	54.2	41.7	50.6	59.8	60.5	63.9	
missing	19.3	10.4	21.2	15.7	8.3	12.5	22.9	8.5	15.8	5.6	
BMI ^2^											b: 0.334; f: 0.445
Normal weight (18.5 to <25.0 kg/m^2^)	26.0	21.8	21.2	16.3	20.8	16.7	24.4	20.8	39.5	34.2	
Overweight (25.0 to <30.0 kg/m^2^)	34.7	38.8	44.2	46.9	37.5	41.7	31.7	36.4	26.3	31.6	
Obesity (≥ 30.0 kg/m^2^)	39.3	39.4	34.6	36.7	41.7	41.7	43.9	42.9	34.2	34.2	
Smoking behavior											b: 0.877; f: 0.882
Smoker	34.2	32.1	28.8	36.5	37.5	45.8	37.8	42.7	31.6	47.4	
Ex-smoker	28.6	23.5	34.6	19.2	20.8	25.0	26.8	23.2	28.9	28.9	
Never-smoker	37.2	44.4	36.5	44.2	41.7	29.2	35.4	34.1	39.5	23.7	
Physical activity (PA)											
PA minutes per week	51 ± 94	98 ± 118	0	0	171 ± 133	33 ± 48	25 ± 57	179 ± 121	99 ± 105	98 ± 94	b:<0.001; f:<0.001
Active days per week	1.3 ± 2.0	2.2 ± 2.3	0	0	3.7 ± 2.1	1.1 ± 1.7	0.7 ± 1.4	3.8 ± 2.0	2.9 ± 2.0	2.5 ± 1.8	b:<0.001; f:<0.001
Chronic diseases ^3^											
Hypertension	25.6	+20.3	22.7	+26.9	30.4	+37.5	28.0	+15.7	20.6	+10.5	b: 0.765; f: 0.027
Diabetes	13.5	+9.6	18.6	+7.7	13.6	+12.5	9.3	+12.0	16.1	+5.3	b: 0.519; f: 0.606
Dyslipidemia	24.7	+17.8	35.6	+25.0	9.1	16.7	22.7	+14.5	25.0	+15.8	b: 0.117; f: 0.456

^1^ Chi-square test for categorical variables, ANOVA for continuous variables. ^2^ Body mass index. ^3^ Self-report of physician’s diagnosis. Baseline: prevalence at baseline. Follow-up: incidence since baseline assessment.

**Table 2 ijerph-17-06349-t002:** Univariable and multivariable logistic regression analysis (complete case analysis, (*n* = 157); **outcome**: stable active trajectory).

Baseline Variables	*n*	Univariable	*p*	Multivariable	*p*
Age (per year)	197	0.99 [0.96; 1.02]	0.516	1.03 [0.97; 1.10]	0.333
Sex	197				
Male vs. female		1.02 [0.49; 2.11]	0.957	0.87 [0.32; 2.39]	0.793
Education	197				
<10 years		Ref.		Ref.	
10–12 years		0.87 [0.36; 2.13]	0.765	0.76 [0.22; 2.63]	0.670
>12 years		2.69 [1.10; 6.61]	0.031	4.20 [1.10; 16.00]	0.036
missing		0.50 [0.06; 4.28]	0.527	n too small	
Household net income	197				
EUR ≤ 1500		Ref.		Ref.	
EUR > 1500		1.61 [0.69; 3.75]	0.275	1.35 [0.38; 4.76]	0.639
missing		1.02 [0.33; 3.14]	0.971	1.42 [0.26; 7.86]	0.689
Migration generation	188				
2nd vs. 1st generation		0.81 [0.33; 2.00]	0.643	0.24 [0.06; 1.05]	0.059
Citizenship	190				
German or dual vs. Turkish		2.4 [1.12; 5.14]	0.024	3.60 [1.20; 10.86]	0.023
Language of questionnaire	197				
German vs. Turkish		2.82 [1.31; 6.07]	0.008	3.35 [1.05; 10.66]	0.041
Marital status	192				
Married (no vs. yes)		1.76 [0.81; 3.80]	0.153	1.50 [0.48; 4.76]	0.487
Smoking behavior	196				
Smoker		Ref.		Ref.	
Ex-smoker		1.12 [0.45; 2.78]	0.806	2.27 [0.61; 8.44]	0.220
Never-smoker		1.19 [0.51; 2.76]	0.693	1.75 [0.55; 5.57]	0.345
BMI ^1^	196				
Normal weight		Ref.		Ref.	
Overweight		0.41 [0.17; 1.02]	0.055	0.28 [0.08; 0.99]	0.047
Obesity		0.49 [0.21; 1.14]	0.097	0.29 [0.08; 1.08]	0.066
Chronic diseases ^2^					
Baseline diagnosis of hypertension (yes vs. no)	176	0.71 [0.29; 1.76]	0.460	0.49 [0.09; 2.57]	0.400
Baseline diagnosis of diabetes (yes vs. no)	171	1.30 [0.44; 3.83]	0.630	1.55 [0.31; 7.70]	0.592
Baseline diagnosis of dyslipidemia (yes vs. no)	174	1.02 [0.42; 2.47]	0.967	1.11 [0.25; 5.00]	0.895

^1^ Body mass index. ^2^ Self-report of physician’s diagnosis. Baseline: prevalence at baseline. Follow-up: incidence since baseline assessment.

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
