# Peer review of "Physical Activity Trajectories among Persons of Turkish Descent Living in Germany—A Cohort Study"

_ijerph, 2020, doi:10.3390/ijerph17176349_

Round 1
Reviewer 1 Report
Abstract
The authors may elaborate on their abstract and provide more critical findings.
Introduction
The introduction section was reasonably performed.
However, the concepts from the authors and the inference of each paragraph may be further addressed. It is also suggested to re-organize the sentences particularly inline 55 to 76 on page 2. It seems the authors roughly provided many findings from previous studies, which should be organized.
Materials and Methods
The physical activity patterns were simply defined according to the differences in weekly PA participating duration (mins per week) between baseline and follow-up (line 109 to 114). It seems incomplete in terms of the “pattern” on physical activity. Specifically, the intensity and types of PA are also critical and important for its outcome. Using the present definition of PA pattern may be limited for further analyses and interpretation, as well as confusing the readers. For example, the “stable active” population might be only showing that they have no variation on PA “duration” within the 6 years, isn’t it? A person who has regular time for PA (e.g. limited time from job, family, etc.) doesn’t mean he/she will do it in the same type or intensity.
I suggest the authors use the idea of “patterns on changes of PA duration” throughout the entire manuscript and its title, but PA patterns instead.
Results
3.2 PA Trajectories among …: What does it mean participants and non-participants? It is not mentioned in the Materials and Methods section, is it? Please mention it clearly, such as sampling, data sources, numbers of the “non-participants”, … everything you have.
Discussion
The discussion section was well presented.
Author Response
Please find our point-by-point response in the attached pdf file.

Reviewer 2 Report
Dear authors,
Thanks for allow me to review the manuscript entitled "Physical activity patterns among persons of Turkish descent living in Germany – a cohort study" which aimed to describe changes of PA over 6 years and to explore migration-related and other predictors of different PA patterns in adults of Turkish descent living in Berlin. I have some concerns about it:
- The introduction should be improved and rewritten. It must clearly show the problem to be treated and justify the need for the study.
Very short and unspecific paragraphs are included. - Sample information should be separate from information regarding study design.
- Only two questions are enough to establish reliable parameters of physical activity? This is the main concern about the study. I need the authors to justify this amply with similar previous studies.
- Write in 1st person should be avoided. Review the entire manuscript.
- Tables' presentation must be improved.
Author Response

(The authors gave the same response as above.)

Reviewer 3 Report
Starting with the title, it is clear, straightforward, and appropriate for the article. The abstract gives a strong overview of the article and goes into some specific methods and results. In the introduction, background information is given on the reasoning of the study, and the purpose is clearly stated. In the methods section, some aspects are thoroughly detailed whereas others are not. The article describes recruitment as a "complex recruitment process via register-based and network approaches using an onomastic procedure" and does not go into further detail on recruitment. I do not believe that the methods are presented in such a way that is easily repeated as they should be. Other than that component of methods, the section is thorough and detailed on measures and statistical analysis. The results section provides all of the pertinent information of the study and uses both tables and graphs to display information. Although the tables have a lot of information, it is understandable and straightforward to read. The graphs assist the reader in understanding the result by showing the trajectory of physical activity over the 6 years. The graphs complement the written results well. All of the discussion is relevant to the study and compares results from there study to similar studies to discuss results. One major weakness is the physical activity assessment. Physical activity was assessed by a few questions that do not go into much detail on the type of physical activity, which could be helpful in analyzing the population and change over time. Another aspect that could be useful in assessing physical activity is to include sedentary activity or work activity
Author Response

(The authors gave the same response as above.)

Round 2
Reviewer 1 Report
Dear authors,
I have seen the great works on your revision, Congratulations!
Reviewer 2 Report
Dear authors,
Thank you for responding satisfactorily to the questions raised.